# FLEXIBLE DIFFUSION FOR GRAPH NEURAL NETWORKS

## ABSTRACT

Graph Neural Networks (GNNs) are attracting growing attention due to their promising performance in modeling a variety of graph-structured data. However, most existing GNNs only consider fixed-range discrete message passing and aggregation, none of them are aware of the importance of the degree and local structure of nodes for smoothing features, which significantly limits the applicability of GNNs. Furthermore, previous approaches either focus on adaptive selection for aggregation structures or treat discrete graph convolution as a continuous diffusion process, lacking holistic consideration of the two issues and resulting in the performance of the model being significantly limited. To this end, we propose a novel Flexible Diffusion Convolution (Flexi-DC), which aims to smooth features by setting a specific continuous diffusion for each node through the degree-and-local structure of the nodes. Specifically, Flexi-DC first extracts the degree and local structure knowledge of the nodes in the graph data and then injects it into the diffusion convolution module to smooth features. Additionally, we also utilize the extracted knowledge for smoothing labels. Flexi-DC is an efficient framework that can significantly improve the performance of most GNN architectures. Experimental results demonstrate that Flexi-DC outperforms their vanilla implementations by an average accuracy of 10.82% (GCN), 12.33% (JKNet), and 11.04% (ARMA) on nine graph datasets with different homophily ratios.

## 1 INTRODUCTION

With the widespread use of graph-structured data, Graph Neural Networks (GNNs) have obtained extensive interest and achieved state-of-the-art graph learning in numerous applications, including image processing (Xing et al., 2021; Adnan et al., 2020; Zorzi et al., 2022), natural language processing (Ma et al., 2021; Chen et al., 2022a; Huang et al., 2020), and recommendation systems (Chen et al., 2022b; Chen and Wong, 2020; Huang et al., 2021). GNNs have the ability to handle data in non-Euclidean structures by learning the representations of target nodes from the neighborhoods and further performing information propagation and prediction on graphs. While their tremendous success in many fields, some issues still need to be addressed. For example, only considering the aggregation of direct neighbors is inconsistent with the structure of graph data, and the discrete aggregation operation limits the information passing between nodes. Recently, many effective improvement methods for the local and discrete aggregation of GNNs have been proposed (Liu et al., 2022; Li et al., 2022; Fey, 2019; Zhao et al., 2021a; Chien et al., 2020), and there is strong evidence that these two limitations can restrict the expressive power and applicability of GNNs.

To tackle the applicability limitation and improve the node representation, two main directions of research are proposed: (1) *Structure-based approaches*: They are based on implementing message passing between global and local neighborhood structures (Abu-El-Haija et al., 2019; Hua et al., 2022; Feng et al., 2022; Wang et al., 2020a; Zhao et al., 2021b); (2) *Attribute-based methods*: Where they make smooth feature propagation based on graph diffusion, comparatively little work has been done to study attribute-based aggregation. The primary researches currently focus on the transformation of aggregated attributes. The purpose of attribute-based aggregation is to enable nodes to capture richer contextual information by extending the discrete messaging process in GCNs to a diffusion process, thereby improving generalization performance. For instance, Graph Diffusion Convolution (GDC) (Gasteiger et al., 2019) constructs a new graph based on a generalized form of graph diffusion, which can be further used to enlarge a larger neighborhood for message passing. Graph Neural Diffusion (GRAND) (Chamberlain et al., 2021) creates multiple different augmented graphs with

node dropping and feature masking, followed by feature propagation. Then the consistency loss is applied to minimize the distances of the representations learned from the augmented graphs.

In this work, to improve the applicability of the model in homophily-heterophily inductive learning tasks, we propose an effective and scalable Flexible Diffusion Convolution based on node degree, namely Flexi-DC. Using the difference in node degrees, which leads the operator to assign different local structures (receptive field) to each node, i.e., nodes with high degrees receive a small receptive field. In contrast, nodes with low degrees receive a relatively larger receptive field. This approach mitigates the limitation of considering only immediate neighbors in previous methods and effectively avoids issues such as over-smoothing and insufficient information aggregation. Meanwhile, we utilize the diffusion kernel function to transform aggregation into continuous form, which boosts the richness and expressiveness of node embedding. In addition, the obtained local structure can also be employed to smooth labels and assist the model to overcome over-prediction and over-fitting problems. Furthermore, the two modules can exploit and enhance each other, ultimately guiding the model to learn information embeddings and improving the applicability of the model. Notably, as a general framework, Flexi-DC can significantly improve vanilla implementations of various popular GNN architectures on multiple real-world datasets with different homophily ratios.

**Summary of Contributions.** (1) We propose a novel framework, called Flexible Diffusion Convolution (Flexi-DC), to improve the predictive performance of the model by flexible diffusion and smoothing label. (2) Degree-based diffusion convolution and label smoothing are developed in our framework. The former leads to more flexible and smooth feature passing, thus improving the expressiveness and applicability of the model, while the latter allows the framework to avoid over-confidence. (3) We provide comprehensive experimental results showing that Flexi-DC can be applied to a variety of GNN models, and it performs considerably better than different vanilla versions and existing SOTA methods across various datasets. Remarkably, the efficiency of the framework is greatly improved over large-scale data datasets.

## 2 PRELIMINARIES

In this section, we first review previous models and based on these models, we will derive our method in the next section. Our work mainly focuses on the problem of semi-supervised node classification. Input an undirected graph network $G = (\mathcal{V}, \mathcal{E})$, where $\mathcal{V}$ denotes the set of vertices with $|\mathcal{V}| = N$ and $\mathcal{E}$ denotes the set of edges, and $\mathbf{A} \in \mathbb{R}^{N \times N}$ refers to adjacency matrix of graph $G$. Given the input feature matrix $\mathbf{X} \in \mathbb{R}^{N \times f}$ with $f$ features for each node and a subset of training node class labels $\mathbf{Y}$ consisting of one-hot label vectors, the task is to predict the labels of the remaining nodes under the supervision of the set of labeled nodes.

**Message-Passing Neural Networks.** Message-Passing Neural Networks (MPNNs) are a class of neural networks that can operate on graph-structured data (Gilmer et al., 2017). In MPNNs, nodes aggregate information from their neighbors and update their own representations based on the received messages (Yadati, 2020). The entire process can be defined as:

$$h_i^{(l+1)} = \psi\left(h_i^{(l)}, \varphi(\{h_j^{(l)} \mid \forall v_j \in \mathcal{N}_i\})\right),$$
(1)

where $h_i^{(l)}$ refers to the hidden feature representation of node $v_i$ in the $l$-th layer. In each layer $l$, the hidden feature matrix of nodes is represented as $\mathbf{H}^{(l)} = [h_1^{(l)}; h_2^{(l)}; \cdots; h_N^{(l)}]$, and $\mathbf{H}$ is initialized by $\mathbf{X}$. $\psi(\cdot)$ refers to the update function and $\varphi(\cdot)$ denotes the aggregation function, while $\mathcal{N}_i$ refers to the aggregated neighbors of node $v_i$. Vanilla GCN (Welling and Kipf, 2016) is an extensively used and popular model of GNNs. It employs a technique known as "renormalization" to add a self-loop to each node in the graph. This technique enables nodes to utilize their own feature information to update their representations and reduce the impact of node degree during the training process. Then, the corresponding updated embedding of GCN is expressed as $\mathbf{H}^{(l+1)} = \sigma(\hat{\mathbf{A}} \mathbf{H}^{(l)} \mathbf{W}^{(l)})$, where $\hat{\mathbf{A}} = \tilde{\mathbf{D}}^{-\frac{1}{2}}(\mathbf{A} + \mathbf{I})\tilde{\mathbf{D}}^{-\frac{1}{2}}$ is a symmetric normalized adjacency matrix with self-connections, and $\tilde{\mathbf{D}}$ is the diagonal degree matrix with $\tilde{\mathbf{D}}_{ii} = \sum_j (\mathbf{A} + \mathbf{I})_{ij}$, $\mathbf{I}$ is the identity matrix.

**Graph Diffusion Convolution.** GNNs typically rely on discrete feature aggregation and propagation operations to update node information, which can lead to loss of embedding information and accuracy. Recently, researchers have focused on extending this discrete operation to continuous

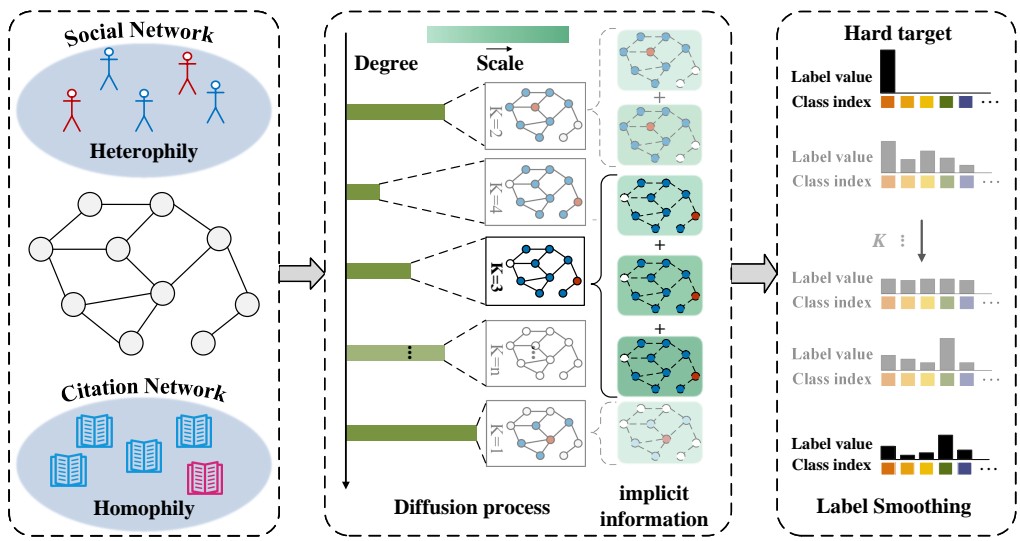

Figure 1: The overall architecture of Flexible Diffusion convolution consists of Degree-based Flexible Diffusion and Label Smoothing, where **K** refers to the **K**-hopping neighbor structure. As is shown in the figure, the leftmost part represents the input of model, which can be either homophily or heterophily; The middle part involves degree-based neighborhood diffusion, which smooths the features and feeds them into an MLP for further processing. Finally, the output is fed into the rightmost part for label smoothing.

graph diffusion (Gasteiger et al., 2019; Zhao et al., 2021a; Xhonneux et al., 2020). The key is that the propagation mechanism of graph diffusion convolution (GDC) addresses this problem in the following way:

$$\psi^{(l+1)}(\hat{\mathbf{H}}, G) = \sum_{k=0}^{\infty} \theta_k \mathbf{T}^k \hat{\mathbf{H}}, \tag{2}$$

where $\hat{\mathbf{H}}$ is the hidden feature matrix of node after aggregating neighbors, $k$ refers to the $k$-th order polynomial in the Laplacian matrix, which depends on nodes within a maximum distance of $k$ steps from the target node. It is worth noting that the summation from 0 to infinity means that each node aggregates information from the entire graph. In Eq. 2, $\mathbf{T}$ is the generalized transfer matrix which is equal to $\hat{\mathbf{A}}$, and $\theta_k$ is the weighting coefficient that satisfies $\sum_{k=0}^{\infty} \theta_k = 1$. Two of the most commonly used diffusion weight coefficients are denoted as the heat kernel ($\theta_k = e^{-t}\frac{t^k}{k!}$) (Kondor and Lafferty, 2002) and personalized PageRank ($\theta_k = \alpha(1-\alpha)^k$) (Brin, 1998).

## 3 FLEXIBLE GRAPH DIFFUSION CONVOLUTION

In essence, Flexible Diffusion Convolution (Flexi-DC) utilizes the continuous local structural information of nodes. The method utilizes kernel functions to continuity for discrete information, which enhances the embedding information of the nodes. The continuity of local structure enhances the flexibility of the model, enabling it to perform feature propagation and aggregation at different distance ranges, thereby better adapting to different graph data and tasks. Since the proposed framework provides a universal continuous aggregation solution and label smoothing techniques, it can easily incorporate different types of GNN models into the framework and be developed and implemented based on the framework. The overall framework is shown in Figure 1.

**Intuition.** The inspiration behind Flexi-GDC is that in graph data, the degree of nodes varies greatly. If we use the same size of local structure (receptive field) for each node in previous GNNs, larger local structures may lead to over-smoothing, while smaller ones may result in instability and insufficient information aggregation, making it difficult to achieve the optimal vector representation

for all nodes. Additionally, because the adjacency matrix is discrete, the local structure is also discrete, making it impossible to describe the continuous relationship between nodes. This limits the information propagation between nodes and may result in information loss or inaccuracies. By designing reasonable aggregation local structure sizes based on node degree information, and using diffusion kernels to reach a continuously changing neighbor radius, the representation of nodes can be made continuous, thereby improving the model's applicability and generalization ability. We break through the limitations of the past, achieving different neighborhood sizes for each node and even different dimensions of the same node, making the model better equipped to handle noise and anomalies in graph data, and improving its performance.

## 3.1 FLEXIBLE DIFFUSION

In graph networks, there exist a significant variation in the number of neighbor nodes, which is referred to as the difference in node degrees. If the traditional GNNs aggregate only the immediate neighbors (1-hop) for message passing, the nodes with higher degrees receive too much information, causing over-smoothing, whereas the nodes with lower degrees may not have enough information for self-enhancement. To address these shortcomings, we propose a method to calculate the appropriate aggregation range based on the degree of each node. The goal of the model is to match the local structure aggregated by each node with the number of its neighbors. We present the details of the operation in Appendix A. By the above operation different local structures are assigned to each node, and then features of the target node are aggregated by equation $\mathbf{H}_i^{(l+1)} = \frac{1}{\mathcal{N}_i} \sum_{k=1}^{\mathcal{N}_i} \left[ (1-\alpha)\hat{\mathbf{A}}^{(k)}\mathbf{H}_i^{(l)} + \alpha\mathbf{H}_i^{(l)} \right]$. In order to take into account the global information of the graph in the model, combining spectral and spatial GNN is proposed to improve the predictive power of the model.

In the spectral GNNs, the graph data is viewed as the discrete signal that can be transformed into frequency domain signal by the Fourier transform. To reduce complexity, Chebyshev $K$-order truncation is used to approximate the convolution kernel, which is expressed as:

$$S' = \sum_{k=0}^{K} \theta_k \mathbf{U} T^k(\tilde{\mathbf{\Lambda}}) \mathbf{U}^T S = \sum_{k=0}^{K} \theta_k T^k(\tilde{\mathbf{L}}) S,$$

where $S'$ is the resulted modulated signal, the Laplacian matrix $\mathbf{L} = \mathbf{D} - \mathbf{A}$ plays a pivotal role (Chung, 1997), normalized to be expressed as $\mathbf{U}(\mathbf{\Lambda})\mathbf{U}^T$, $\mathbf{U}$ is the eigenvector matrix and $\mathbf{\Lambda}$ is the diagonal eigenvalue matrix of $\mathbf{L}$. Note that it has meaningful guidance for any given vector, eg. $\mathbf{H}^T \mathbf{L} \mathbf{H} = -\sum_{v,j\in\mathcal{E}} (h_i - h_j)$. It can be easily demonstrated that, for a square lattice in Euclidean space of $m$-dimensional, with grid spacing $\eta$, $\mathbf{L}/\eta$ is nothing but a finite difference approximation of the well-known continuous Laplacian (Kondor and Lafferty, 2002):

$$\Delta = \frac{\partial^2}{\partial x_1^2} + \frac{\partial^2}{\partial x_2^2} + \cdots + \frac{\partial^2}{\partial x_m^2},$$

as $\eta \to 0$, this approximation achieves heightened precision. Following the principles of classical physics, the equation takes on the following format: $\frac{d}{d_t}\xi = \mu \Delta \xi$ is utilized to describe the diffusion of heat and other substances within a continuous medium. The diffusion equation employed within graph neural networks is denoted by:

$$\frac{d}{d_\beta} \mathbf{Kel}_\beta = \mathbf{L} \, \mathbf{Kel}_\beta. \tag{3}$$

The presentation on diffusion kernel is presented in Appendix A. To put it succinctly, we can achieve the Flexible Diffusion Convolution (Flexi-DC) by extending the feature propagation function present in Eq. 3 to each layer, node, and channel:

$$\mathbf{H}_i^{(l+1)} = \sigma \sum_{k=0}^{K} \left( \mathbf{Kel}_{\beta_i}^k \, \mathbf{H}_i^{(l)} \right) = \sigma \sum_{k=0}^{K} \left( \frac{\lambda^k e^{-\lambda}}{k!} \, \mathbf{H}_i^{(l)} \right). \tag{4}$$

We present the settings of the parameters in the flexible diffusion convolution in Appendix C.

## 3.2 CONTINUITY TRAINING

The diffusion convolution allows us to use continuous kernel functions in place of the discrete feature aggregation functions of GNNs. Considering the different radii of neighbors between different nodes in different layers, we design a flexible local structure selection based on node degree and then use kernel functions to continuously transform the adjacency matrix. By doing so, we can replace the manual adjustment of the neighbor radius and achieve optimal prediction accuracy by updating the kernel radius via gradient descent.

To prevent over-fitting of the model, we propose a bi-level optimization strategy to help the model achieve better performance. Specifically, we first train the model weights on the training set and then find the optimal kernel radius based on the minimum loss on the validation set using the optimal weights (Zhao et al., 2021a; Colson et al., 2007).

$$w^* = \arg \min_{w} \mathcal{L}_{train}(\lambda, w), \tag{5}$$

$$\lambda^* = \arg \min_{\lambda} \mathcal{L}_{val}(\lambda, w^*). \tag{6}$$

To find the optimum $\lambda$, we need to find the optimal convergence value of $w$ before updating the search for $\lambda$ at each epoch, which adds significant complexity to the entire model training process. Inspired by neural architecture search (Liu et al., 2018) and gradient-based hyperparameter tuning (Luketina et al., 2016), we use an approximation method to update $\lambda$ at the same time as $w$ when updating the weights. To reduce the time consumption brought by the two-stage optimization, we put the degree-based feature aggregation enhancement in Section 3.1 outside of the model training.

## 3.3 LABEL SMOOTHING

For training GNNs, most existing work focuses on minimizing the expected cross-entropy loss between the ground truth targets $T(c \mid i)$ and the prediction of the model $P(c \mid i)$ of the model. Minimizing this loss is equivalent to maximizing the expected log-likelihood of the labels selected according to the distribution $T(c \mid i)$. However, using hard target ($T(y_i \mid i) = 1$ and $T(c \mid i) = 0$, $\forall c \in y_i$) to train GNNs ignores the correlation between connected nodes. When classifying node $i$, the neighbor features are first aggregated through message passing, but node $j \in \mathcal{N}_i$ may not belong to any class other than class $y_i$. Forcing the connected nodes of different classes to be assigned the full probability as predictions ignore their structural relationship.

To overcome the over-confidence issue mentioned above, label smoothing has been proposed to find a distribution. Specifically, we use the labels of node neighbors to explore features beyond its own hard target, and use structural information to generate soft targets for adaptive graph structures. The method for smoothing label is defined as follows:

$$P^{SL} = \frac{1}{|\mathcal{N}_i|} \sum_{k=1}^{\mathcal{K}_i} \left[ (1 - \alpha) \, \hat{\mathbf{A}}^{(k)} P_i + \alpha P_i \right], \tag{7}$$

where $\mathcal{K}_i$ follows the definition in Eq. 8, and $\alpha$ is smoothing coefficient. Thus, each node takes into account the impact of its connected nodes on itself before proceeding with the classification. Details regarding the hyperparameters and some aspects of the model can be found in Appendix C.

## 4 EXPERIMENTS

In this section, we evaluate Flexible Diffusion Convolution on nine real-world node-level datasets. Specifically, we primarily 1) empirically validate that Flexi-DC achieves SOTA performance on multiple datasets; 2) demonstrate the applicability of the framework in heterophilous networks. Furthermore, we analyze the running time of Flexi-DC is described in Appendix B.3.

## 4.1 EXPERIMENTAL SETUP

To ensure the accuracy and reliability of our results, we have taken several measures in our experiments. We followed standard procedures and reported the results on an average of 100 random data splits and initializations. All baseline GNNs use the same hyperparameters as the baseline model

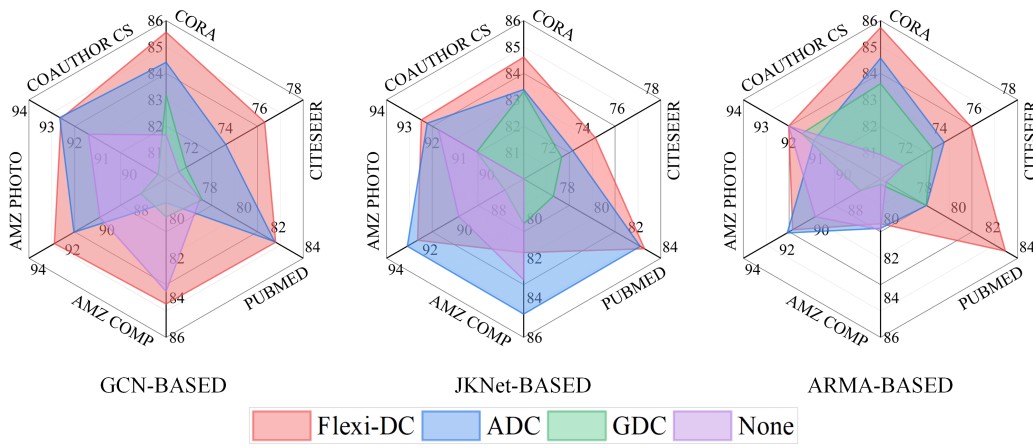

Figure 2: The accuracy of node classification (%), with different comparison frameworks based on GCN, JKNet, and ARMA. Flexi-DC exhibits superior performance over other competing models in a multi-dimensional way. Each dataset corresponds to its nearest coordinate axis.

of AD (Zhao et al., 2021a). We set the learning rate of $t$ to be the same as the learning rate of other parameters, and the specific setting will be provided in Appendix C. All results are reported in the form of averages, along with 95% confidence intervals calculated using bootstrapping.

**Datasets.** The prediction task is focused on semi-supervised node classification using widely adopted datasets, including CORA (McCallum et al., 2000), CiteSeer (Sen et al., 2008), PubMed (Namata et al., 2012), Amazon Computers, Amazon Photo, Coauthor CS (Shchur et al., 2018), Cornell, Texas, and Wisconsin (Pei et al., 2020). Details of the dataset are presented in Appendix B.1. We only use the largest connected component in these datasets and split them into development and test sets. The development set contains 1500 nodes, with 20 nodes per class, except for the Coauthor CS dataset, and the remaining nodes are used as the validation set.

**Baselines.** We also implement Flexi-DC on three models: GCN (Kipf and Welling, 2016), JKNet (Xu et al., 2018b), and ARMA (Bianchi et al., 2021). We replace the original feature aggregation function with Flexi-DC while retaining the feature transformation function, and applied label smoothing technique on the central node's neighborhood range. We set the expansion step ($K$ in Eq. 4) to 10, and the selection of values for $K$ is discussed in Section 4.4. We use early stopping with patience of 100 epochs to effectively improve the algorithm's generalization across different datasets.

## 4.2 RESULT AND ANALYSIS

Figure 2 displays the performance of GDC, ADC, and the proposed Flexi-DC on six datasets using GCN, JKNet, and ARMA. The results indicate that Flexi-DC significantly improves upon the basic GNN and outperforms ADC and GDC on most datasets, suggesting that our proposed method is robust and generalizable for these types of problems.

Furthermore, on some larger datasets, although the performance of Flexi-DC does not reach the optimal level and is even slightly lower than that of other compared algorithms, all GPU execution time analyses in Appendix B.3 demonstrate that our proposed method consumes far less time than ADC. The reason for the lack of performance improvement may be that the scale of the dataset is too large, and the selection of local structures does not significantly enhance the optimal representation vector of the target node. To further investigate this phenomenon, we plan to conduct more in-depth analyses and research on these larger datasets in future work. The distribution of experimental results for 100 runs is presented in Appendix B.2.

In summary, the experimental results demonstrate that our proposed Flexi-DC has competitive performance on multiple datasets based on different basic GNNs and outperforms other competing methods in most cases.

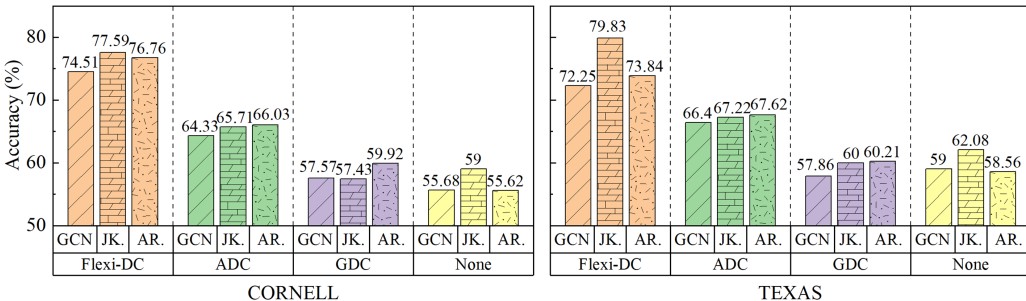

Figure 3: Algorithmic comparison of node classification accuracy on Cornell and Texas datasets, where JK., AR. refer to JKNet and ARMA, respectively.

### 4.3 APPLICABILITY ANALYSIS

To verify the applicability of the proposed framework, we conducte experiments on heterophilic graphs. Figure 3 displays the results of various frameworks based on three fundamental GNNs (GCN, JKNet, and ARMA) on the CORNELL and TEXAS datasets. Since nodes with different labels are more likely to be connected in heterophilic graphs, considering an excessive number of neighbors can adversely affect the generation of optimal node embeddings. Hence, the choice of $K$ is smaller compared to Section 4.2. We place the results of other datasets in Appendix B.4, and the outcomes are consistent with those in CORNELL and TEXAS, outperforming competing models.

As shown in Figure 3, Flexi-DC achieves the best performance across different GNNs, with average performance enhancements of 14.47%, 28.85%, and 29.97% compared to ADC, respectively. This suggests that Flexi-DC exhibits robustness and broad applicability when processing these heterophilic graphs. Furthermore, the computational efficiency of Flexi-DC in each heterophilous graph is presented in Appendix B.3. We notice that when the framework is based on JKNet, the computation time is nearly identical to ADC, even less on larger datasets, and remains similar on other datasets, further substantiating its practicality in these graph data. This is primarily attributed to conducting continuous aggregation operations of target nodes within a smaller local scope, considering hidden information within the local structural range while avoiding noise introduced by larger structures.

In conclusion, the analysis of experimental results in the bar chart demonstrates that our proposed idea significantly enhances Flexi-DC's applicability.

### 4.4 SENSITIVITY ANALYSIS

Consider whether larger local structures are really useful for achieving the best representation vectors of nodes. In Eq. 4, $K$ represents the number of steps in the Taylor expansion or the flexible neighborhood structure of nodes. Due to the different structures of homophilous and heterophilous graphs, the choice of $K$ may vary, and we can improve the applicability of the model by choosing the appropriate number of steps. Moreover, by changing $K$, we can check whether neighbors beyond $K$ steps are truly important. Figure 4 shows the demand for the number of Taylor expansion steps.

The experimental results indicate that increasing the number of Taylor expansion steps generally improves the accuracy in homophilous graphs, but the accuracy no longer improves beyond ten steps. This suggests that aggregating the neighbors of the center node within a certain local structure can improve its representation, but when the local structure becomes too large, over-smoothing may occur, leading to diminished information value. In heterophilous graphs, the labels of neighbor nodes around the center node are usually different from their own, and considering too large local structures may introduce noise and affect the representation vector of the center node. Therefore, we generally consider only the direct neighbors. We improve the model's expressiveness by performing continuous aggregation, which considers the hidden features between nodes during the discrete operation.

Overall, our degree-based aggregation method can help the model find appropriate local structures for different datasets, while continuous aggregation enhances feature representation.

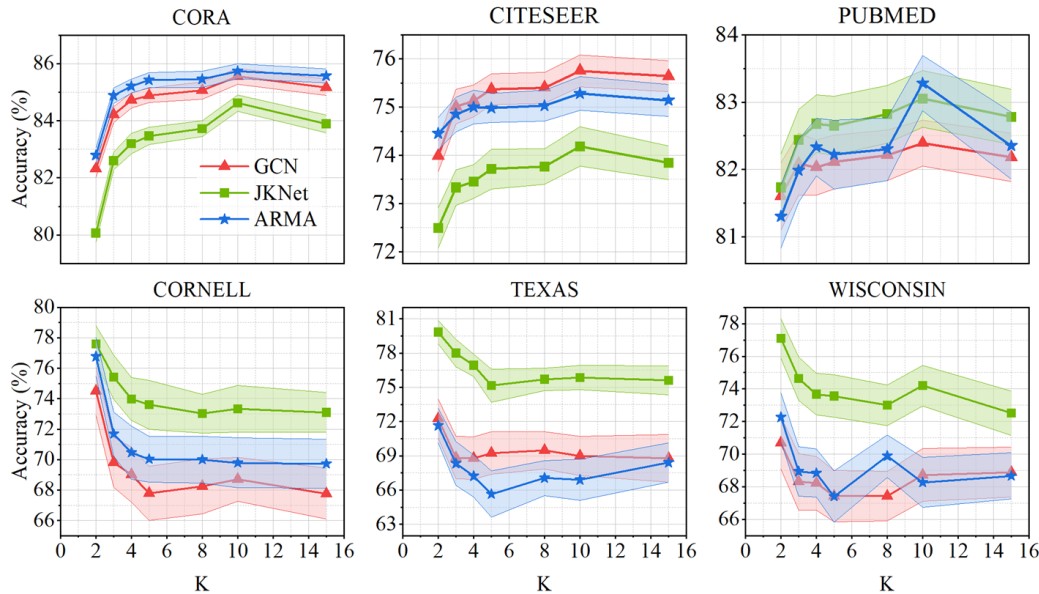

Figure 4: The influence of the Taylor expansion steps (local neighborhood structure) on the performance of Flexi-DC. In most cases of homophilous graphs, increasing the expansion order helps to improve accuracy, while the opposite is true for heterophilous graphs.

Table 1: The classification accuracy (%) ablation studies for two sub-modules, where Flexible-DC w/o Diffusion, w/o SL and w/o flexible refer to the absence of label smoothing, flexible convolution and node-wise receptive field modules, respectively.

| Method | Cora | Cite. | Pubm. | Comp. | Phot. | CS | Corn. | Texas | Wisc. |
|---|---|---|---|---|---|---|---|---|---|
| Flexi-DC (GCN) | **85.56** | **75.75** | **82.39** | **84.31** | **92.52** | 92.83 | **74.51** | **72.25** | **78.68** |
| Flexi-DC w/o Diffusion | 81.86 | 63.58 | 76.97 | 83.15 | 89.64 | 92.26 | 67.16 | 69.14 | 70.69 |
| Flexi-DC w/o SL | 84.75 | 75.39 | 81.87 | 83.50 | 92.49 | 92.61 | 73.90 | 71.97 | 70.34 |
| Flexi-DC w/o Flexible | 83.74 | 72.68 | 80.86 | 83.81 | 92.50 | **92.97** | 67.48 | 70.56 | 73.91 |
| Flexi-DC (JKNet) | **84.62** | **74.18** | **83.05** | 81.72 | **92.19** | 92.73 | **77.59** | **79.83** | **80.90** |
| Flexi-DC w/o Diffusion | 81.11 | 69.06 | 79.28 | 80.68 | 89.74 | 91.51 | 72.46 | 72.83 | 77.09 |
| Flexi-DC w/o SL | 83.24 | 73.44 | 82.38 | 79.51 | 91.38 | 92.68 | 77.18 | 79.59 | 76.33 |
| Flexi-DC w/o Flexible | 83.61 | 73.04 | 82.96 | **82.41** | 91.38 | 92.31 | 70.86 | 71.43 | 78.52 |
| Flexi-DC (ARMA) | **85.74** | **75.28** | **83.28** | 80.26 | 91.13 | **92.35** | **76.76** | **73.84** | **80.78** |
| Flexi-DC w/o Diffusion | 83.15 | 72.16 | 80.13 | **83.32** | **92.54** | 92.20 | 73.70 | 71.63 | 72.26 |
| Flexi-DC w/o SL | 85.18 | 74.64 | 81.72 | 77.81 | 90.15 | 93.33 | 75.44 | 70.68 | 71.60 |
| Flexi-DC w/o Flexible | 84.05 | 73.85 | 81.16 | 82.81 | 91.93 | 92.22 | 72.78 | 72.25 | 77.01 |

## 4.5 ABLATION STUDY

For comparison purposes, the share GNN with the parameters shown in Section 4.4 is adopted by default in this paper. In this subsection, we evaluate and validate our proposed Flexi-DC through ablation experiments. Specifically, we compare the importance and effectiveness of different components of the algorithm on nine datasets based on the vanilla GNN architecture.

From Table 1, we can observe that the impact of the different components of our model on overall performance is varied. Concretely, we find that the flexible diffusion convolution component has a more significant impact on the performance of our model, while the impact of the label smoothing component is relatively minor. This may be attributed to the fact that flexible diffusion convolution considers both hidden and explicit features of the nodes, thereby enhancing the expressive power of the nodes, while the primary purpose of smoothing the label is to prevent the model from being

over-confident, making it a less crucial component of the framework. However, these experimental results indicate that all the modules have achieved the positive effects we expected.

In conclusion, the experimental results also demonstrate the importance of flexible diffusion convolution to the framework, further indicating that the use of both hidden and explicit node features can significantly improve downstream task prediction results.

## 5 RELATED WORK

**Graph Neural Networks.** GNNs have garnered significant attention due to their ability to achieve competitive performance on various tasks (Kipf and Welling, 2016). GNNs could be divided into spectral domain and spatial domain models. The spectral GNNs (Bruna et al., 2013; Balcilar et al., 2021; Wang and Zhang, 2022) represent graph data as eigenvectors of its corresponding graph Laplacian matrix, allowing for analysis and processing of graph data. Spatial GNNs operate directly on the nodes to extract and aggregate features using convolution and pooling layers, simplifying spectral GNNs, and providing better interpretability, higher computational efficiency, and wider applicability to complex graph structures. Spatial GNNs for instance GAT (Veličković et al., 2017), GCN (Kipf and Welling, 2016), GIN (Xu et al., 2018a), and GraphSAGE (Hamilton et al., 2017) can also be understood in terms of message passing.

**Graph Augmented Learning.** There are some works (Zhu et al., 2020; Wang and Derr, 2021; Chen et al., 2020; Chien et al., 2020; Chen et al., 2013; Cai et al., 2021; Zhao et al., 2021a; Park et al., 2022) improve the representation of nodes through graph augmentation techniques. For example, $H_2$GCN (Zhu et al., 2020) proposes aggregating information from higher-order neighbors at each message passing step. TDGNN (Wang and Derr, 2021) utilizes tree decomposition to separate neighbors of different $K$-hops into multiple subgraphs, and then parallelizes message propagation on these subgraphs. GPR-GNN (Chien et al., 2020) further assigns learnable weights and adaptively performs graph convolution of each layer by the Generalized PageRank (GPR) technique. ADC (Zhao et al., 2021a) supports automatic learning of the optimal neighborhood from the data, thereby eliminating the manual search process for the optimal propagation neighborhood in GDC. However, these considerations are somewhat one-sided and lack a combined reflection on local structure and diffusion aggregation, which makes them not applicable to heterophilous graphs.

**Label Smoothing.** Label smoothing is a regularization technique designed to alleviate overfitting and improve model generalization in deep neural network (Jia and Benson, 2020; Szegedy et al., 2016; Zhang et al., 2021). This method computes a weighted average of the original (one-hot) label and the labels of neighboring nodes to generate a new smoothed label, reducing noise and uncertainty during training. Wang et al. (Wang et al., 2020b) propose a hierarchical label smoothing method, which imposes a higher smoothing penalty on predictions with low confidence compared to those with high confidence. Ghoshal et al. (Ghoshal et al., 2021) present a low-rank adaptive label smoothing method that improves deep neural network training by adapting to the latent structure of the label space in structured prediction tasks.

Different from the above-mentioned model augmentation techniques, our goal is to construe a neighborhood structure based on node degree to guide the augmentation of node representation.

## 6 CONCLUSION

In this work, we address the problem of the applicability of GNNs. In contrast to existing methods, we propose a Flexible Diffusion Convolution (Flexi-DC) algorithm, which integrates diffusion convolution and label smoothing to balance information learning through node degree and local structures. Specifically, Flexi-DC first designs an operator for the degree and local structure of nodes, and then assigns a neighborhood range for each node based on the operator. By this means, Flexi-DC can distribute different receptive fields of varying sizes to each node, thereby avoiding the problems of over-smoothing and insufficient information aggregation. Moreover, Flexi-DC improves the applicability of the model by utilizing flexible neighborhood ranges for smoothing labels. We conduct experiments on nine real-world datasets. The empirical results verify that Flexi-DC achieves outstanding performance on various graph datasets and for different GNN backbones, exhibiting better applicability and higher efficiency.

## REPRODUCIBILITY STATEMENT

**(This section does not count towards the page limit.)**

We provide the detailed algorithm description and experimental implementation details in Appendix C. We will make our codes and pre-trained checkpoints publicly available to facilitate the replication and verification of our results upon publication.

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
