APPENDIX OF PAPER "FLEXIBLE DIFFUSION FOR GRAPH NEURAL NETWORKS"

APPENDIX

## A  ADDITIONAL DESCRIPTION OF FLEXI-DC

Message Passing Neural Networks, typically called spatial graph neural networks, consider only their direct neighbors at each layer of aggregation, and the processing ability for global features is relatively weak. On the other hand, spectral-based graph neural networks do not just rely on first-hop neighbors and capture more complex graph properties, which can learn the global features of the graph in the spectral domain. However, these methods are usually outperformed by MPNN on graph-related tasks (Veličković et al., 2017; Xu et al., 2018a), and cannot be generalized to previously unseen graphs because of the computational complexity. Therefore, we improve the learning of global and local features by combining the spectral and spatial domains to promote the applicability of the model. Source code is available at `https://anonymous.4open.science/r/Flexi-DC-4965/`.

**Degree-based Neighborhood.** By normalizing the degree, we scale the data values to a fixed range and assign larger local structures to nodes with higher degrees and vice versa. We propose the following computation method for assigning the neighborhood range based on the degree of each node:

$$\mathcal{K}_i = \mathcal{K}_{min} + (\mathcal{K}_{max} - \mathcal{K}_{min}) \frac{log(degree_{max} + 1) - log(degree_{v_i} + 1)}{log(degree_{max} + 1) - log(degree_{min} + 1)}, \tag{8}$$

where $\mathcal{K}_{min}$ and $\mathcal{K}_{max}$ respectively denote the minimal and maximal neighborhood structures in terms of allocation, while $degree_{max}$ and $degree_{min}$ represent the maximum and minimum degrees of nodes in the dataset. This process satisfies the aforementioned requirement: nodes with larger degrees are allocated relatively fewer neighbors. By assigning different local structures to nodes based on their degrees, we can meet the information demands of each node and achieve optimal vector representations.

**Diffusion Kernel.** The heat kernel is a fundamental concept in mathematics and physics, particularly in the study of partial differential equations and stochastic processes. It is a solution to the heat equation, which describes how heat diffuses over time. The heat kernel describes the probability distribution of the locations of particles undergoing Brownian motion or other diffusive processes. It is also closely related to the concept of Laplace operator, which is a fundamental operator used in the study of geometry and topology. The feature propagation speed between two nodes is proportional to the difference between their features. Formally, this prior knowledge can be described as:

$$\frac{dx_i(t)}{dt} = - \sum_{v_j \in \mathcal{N}_i} \tilde{A}_{ij} \left( x_i(t) - x_j(t) \right) \tag{9}$$

This differential equation can be solved as:

$$\mathbf{X}(t) = \mathbf{H}_t \mathbf{X}(0) \tag{10}$$

where $\mathbf{X}(t)$ is the feature matrix after diffusion time $t$ and $\mathbf{H}_t$ is diffusion or heat kernel with expression $e^{-(\mathbf{I}-\mathbf{T})t}$. In graph neural networks, diffusion kernel can be defined by an exponential function of the Laplacian matrix to describe the similarity and distance between nodes. First, we use the Taylor expansion to obtain an explicit $k$-order polynomial approximation for the diffusion kernel, that is,

$$e^{-(\mathbf{I}-\mathbf{T})t} = \sum_{k=0}^{\infty} e^{(-t)} \frac{t^k}{k!} \mathbf{T}^k$$

## B  ADDITIONAL EXPERIMENTS

### B.1  EXPERIMENTAL ENVIRONMENT

Many existing works on graph modeling have an underlying assumption of network homophily, which limits their ability to address network heterophily/discrepancy, where connections tend to exist

Table 2: Benchmark dataset statistics.

| Dataset | #Nodes | #Edges | #Features | #Classes | #Homophily |
| --- | --- | --- | --- | --- | --- |
| Cora | 2708 | 5278 | 1433 | 7 | 0.825 |
| Citeseer | 3327 | 4552 | 3703 | 6 | 0.718 |
| Pubmed | 19717 | 44324 | 500 | 3 | 0.792 |
| Amazon Computers | 13752 | 245861 | 767 | 10 | 0.802 |
| Amazon Photo | 7650 | 119081 | 745 | 8 | 0.849 |
| Coauthor CS | 18333 | 81894 | 6805 | 15 | 0.832 |
| Cornell | 183 | 295 | 1703 | 5 | 0.301 |
| Texas | 183 | 309 | 1703 | 5 | 0.057 |
| Wisconsin | 251 | 499 | 1703 | 5 | 0.213 |

between different types of nodes. In contrast, we want to improve the applicability of the model in both homophilous and heterophilous networks through flexible diffusion convolution.

**Homophily.** Homophily reflects the preference of nodes to select neighbors. For a strongly Homophilous graph, nodes tend to form connections with nodes that have the same label. The homophily ratio $h$ measures the level of overall homophily in the graph. Several homophily metrics have been proposed with different focuses (Lim et al., 2021; Pei et al., 2020). We adopt edge homophily as defined in (Zhu et al., 2020), which is expressed as:

$$ h = \frac{\left|\{(v_i, \ v_j) : (v_i, \ v_j) \in \mathcal{E} \wedge Y_{v_i} = Y_{v_j}\}\right|}{|\mathcal{E}|} $$

It is evident that when $h$ is large ($h \rightarrow 1$), the graph has high homophily, and when $h$ is small ($h \rightarrow 0$), the graph has high heterophily (low homophily).

**Datasets.** Cora, Citeseer, and Pubmed (Sen et al., 2008) are standard citation graphs where nodes denote papers while the bag-of-words of papers are used as node features. The edges of the graph represent citations between papers, and the labels of the nodes refer to the research topic. Moreover, the three datasets are homophilous graphs. Amazon Computers and Amazon Photo are fragments of the Amazon co-purchase graph (McAuley et al., 2015), where nodes represent products and edges indicate frequent co-purchase of two products. Node features are bag-of-words encoded product reviews, and class labels are given by the product category. Coauthor CS is a co-authorship graph based on the Microsoft Academic Graph for the 2016 KDD Cup Challenge[1]. Here, nodes represent authors and are connected by an edge if they co-authored a paper. Node features represent the paper keywords for each author, while class labels indicate the most active research field for each author. WebKB[2] is a web dataset collected from computer science departments of various universities. In this dataset, nodes represent web pages, and edges represent hyperlinks between web pages. Node features are represented by the bag-of-words for each web page. Specifically, Texas, Cornell, and Wisconsin are selected as benchmark experiments. The web pages in the dataset are divided into five categories: students, projects, courses, staff, and faculty. Table 2 summarizes their statistics.

### B.2 ADDITIONAL RESULT ANALYSIS

In this experiment, we recorded the experimental results by running the experiments 100 times. To better illustrate the robustness of the proposed framework, we plotted the results as box plots. They are useful in providing a summary of the distribution of the data, including the median, quartiles, and outliers. The box in the plot represents the interquartile range (IQR), which spans from the first to the third quartile. The median is indicated by a horizontal line within the box. The whiskers extend from the box to the minimum and maximum values within 1.5 times the IQR. Any data points that fall outside of the whiskers are considered outliers and are plotted as individual points.

In Figure 5, we observe that the proposed framework achieves a relatively stable data distribution across all vanilla GNNs on which it is based, with higher maximum accuracy obtained compared to

---

[1]https://kddcup2016.azurewebsites.net/.

[2]http://www.cs.cmu.edu/afs/cs.cmu.edu/project/theo-11/www/wwkb/.

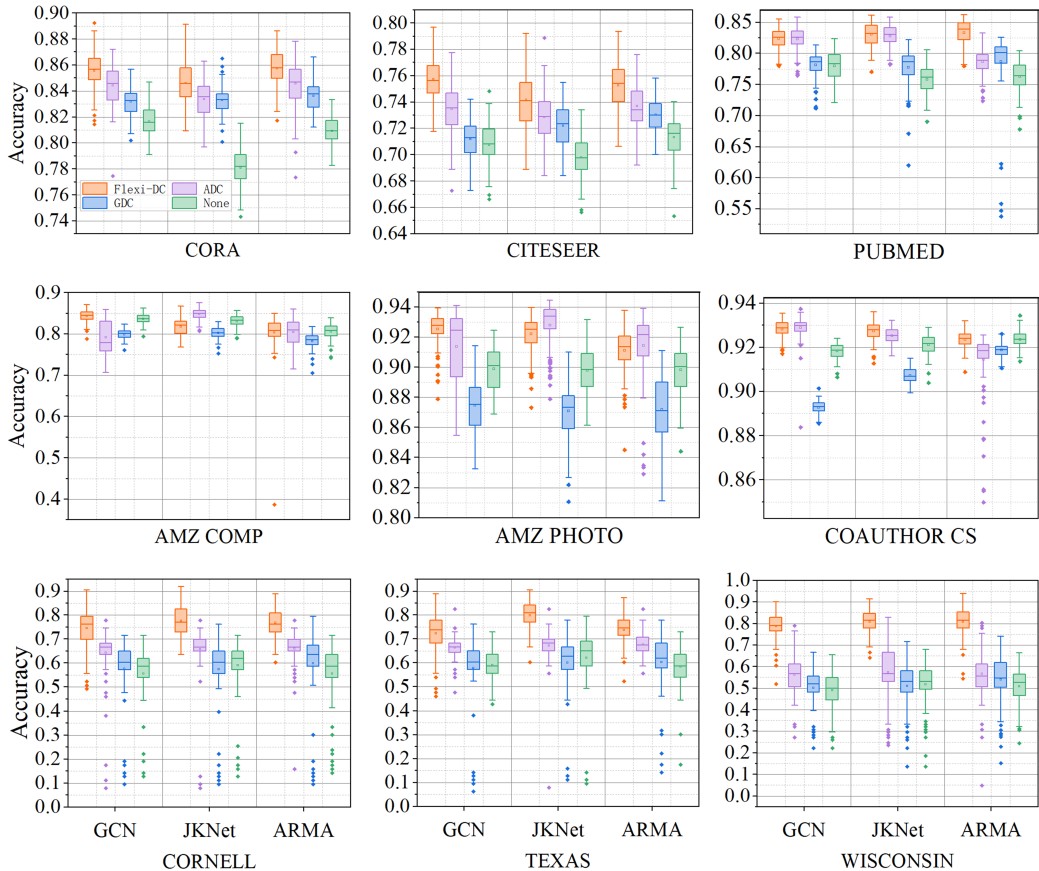

Figure 5: Distribution of results for different methods run 100 times on different datasets.

the baseline versions and very few outliers. Moreover, we find that the distribution of the compared competing algorithms varies significantly across different datasets, with some datasets having a more uniform distribution while others having a highly imbalanced distribution with many outliers, demonstrating the instability of these algorithms. Our proposed framework effectively avoids such situations and significantly reduces the frequency of outliers, resulting in less variability in the results.

The experimental results on different datasets are also presented in Table 3, where our framework achieves relatively outstanding performance. In particular, in the heterogeneous matching network, the overall performance improves by **20.56** *vs.* ADC (GCN), **33.75** *vs.* GDC (GCN), **25.24** *vs.* ADC (JKNet), **37.93** *vs.* GDV (JKNet), and **21.63** *vs.* ADC (ARMA), **40.24** *vs.* GDC (ARMA). Although the desired results are not obtained in some large-scale data, the GPU time consumption is greatly reduced, as will be presented in Appendix B.3.

## B.3    RUNNING TIME

We compare the running time of each method based on different vanilla GNNs on all the datasets. We use the early stopping strategy of 300 epochs. Figure 6 to 8 show the running time of all the methods. From Figure 6, we can observe that versions Flexi-DC and ADC based on GCN have similar GPU running times on small-scale datasets. The reason why our method does not have an advantage is that the small dataset size required us to generate rich hidden feature information through flexible diffusion so that the node representations could be improved and downstream task prediction accuracy could be promoted. However, in large-scale graphs, our proposed method significantly reduces time consumption due to the enhanced technique, which reduces the node aggregation time consumption. From a global perspective, our model has a much lower average time on all datasets compared to the

Table 3: The classification accuracy (%) across all methods on nine datasets: "average accuracy ± 95% confidence intervals calculated via bootstrapping". The error (±) represents the 95% confidence intervals calculated via bootstrapping of the results obtained for each method run for 100 tests. We highlight the optimal results ( average accuracy) for different vanilla GNNs on each data set.

| Datasets | Flexi-DC | | | ADC | | | GDC | | |
| --- | --- | --- | --- | --- | --- | --- | --- | --- | --- |
| | GCN | JKNet | ARMA | GCN | JKNet | ARMA | GCN | JKNet | ARMA |
| Cora | 85.56±0.28 | 84.62±0.29 | 85.74±0.26 | 84.42±0.29 | 83.39±0.29 | 84.58±0.36 | 83.19±0.22 | 83.31±0.22 | 83.61±0.22 |
| Cite. | 75.75±0.33 | 74.18±0.41 | 75.28±0.35 | 73.48±0.38 | 72.86±0.40 | 73.71±0.32 | 71.17±0.30 | 72.21±0.31 | 73.05±0.28 |
| Pubm. | 82.39±0.34 | 83.05±0.42 | 83.28±0.41 | 82.33±0.38 | 82.80±0.34 | 78.68±0.40 | 78.10±0.39 | 77.73±0.62 | 78.66±1.06 |
| Comp. | 84.31±0.29 | 81.72±0.27 | 80.26±0.92 | 79.20±0.81 | 84.82±0.29 | 80.49±0.61 | 79.91±0.21 | 80.26±0.28 | 78.27±0.35 |
| Phot. | 92.52±0.23 | 92.19±0.23 | 91.13±0.30 | 91.38±0.45 | 92.79±0.30 | 91.45±0.45 | 87.47±0.38 | 87.10±0.36 | 87.17±0.45 |
| CS | 92.83±0.07 | 92.73±0.08 | 92.35±0.07 | 92.88±0.12 | 92.52±0.07 | 91.43±0.28 | 89.29±0.06 | 90.71±0.07 | 91.87±0.06 |
| Corn. | 74.51±1.60 | 77.59±1.22 | 76.76±1.27 | 64.33±2.33 | 65.71±2.30 | 66.03±1.52 | 55.68±2.38 | 59.00±2.38 | 55.62±2.65 |
| Texas | 72.25±1.73 | 79.83±1.03 | 73.84±1.62 | 66.40±0.97 | 67.22±1.65 | 67.62±0.87 | 59.00±1.10 | 62.08±2.84 | 58.56±1.63 |
| Wisc. | 78.68±1.25 | 80.90±1.01 | 80.78±1.28 | 56.27±1.86 | 57.36±2.67 | 56.58±2.22 | 48.98±1.68 | 51.70±2.15 | 50.81±1.68 |

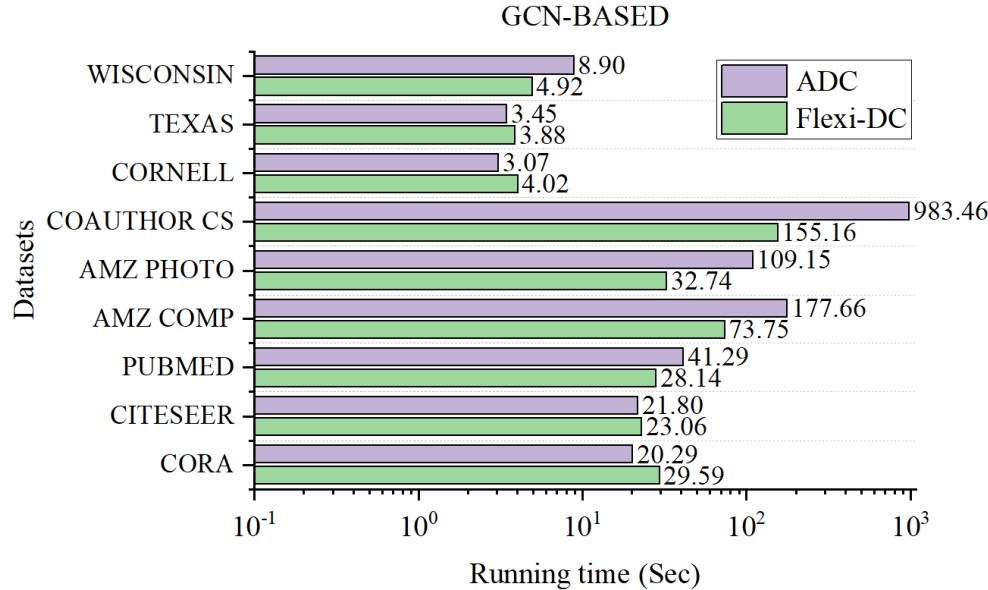

Figure 6: Comparison of the running time of GCN-based methods.

comparative models. This phenomenon is also observed in the JKNet- and ARMA-based models, as shown in Figures 7 and 8, respectively.

## B.4 ADDITIONAL APPLICABILITY ANALYSIS

This part supplements Section 4.3, with the experimental settings kept consistent. We add experimental results to the WISCONSIN dataset. As shown in Figure 9, our proposed framework achieves the best performance across different vanilla GNNs and an overall average performance improvement of 41.21% over the second-best method (ADC).

The experimental results demonstrate that our proposed flexible diffusion convolution and label smoothing can effectively improve the adaptability of the model. This is mainly attributed to our continuous aggregation operation of the target node in a small local range, which considers the hidden information within the local structural range and avoids the noise introduced by larger structures. Previous works have neglected the hidden information of nodes in graph data and only aggregated the explicit information of nodes, leading to performance limitations.

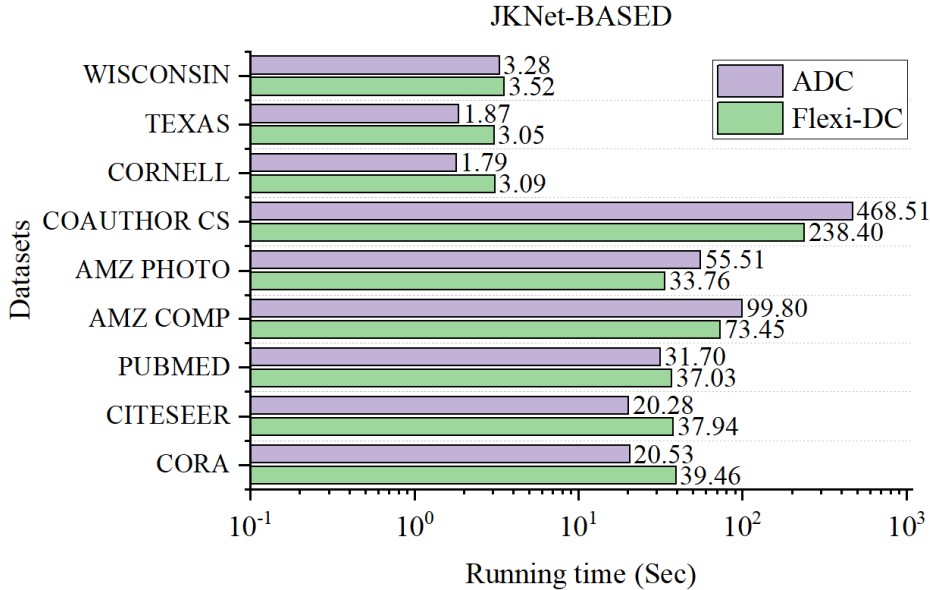

Figure 7: Comparison of the running time of JKNet-based methods.

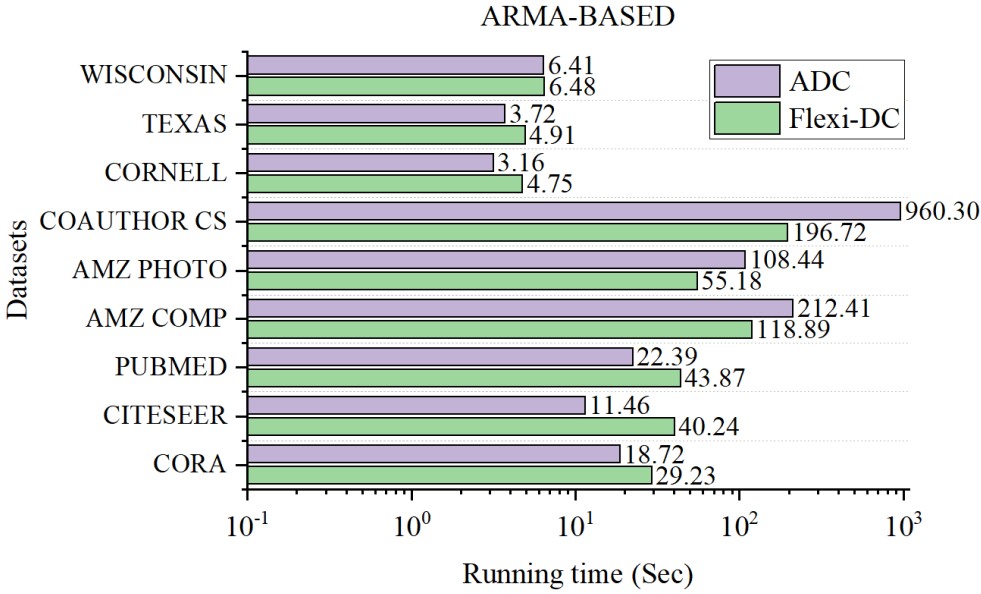

Figure 8: Comparison of the running time of ARMA-based methods.

## B.5 PERFORMANCE DEFICIENCY ANALYSIS

In large-scale graph data, the size and density of the graph can be enormous. Aggregating neighboring nodes using the graph structure throughout the entire graph already provides much neighborhood information. However, considering the hidden information of nodes through diffusion convolution may lead to over-smoothing issues, which can result in poor performance, as shown in Section 4.2.

We conducted sensitivity experiments on our proposed framework on large-scale datasets to verify the explanation of over-smoothing mentioned above. Consider the order of Taylor expansion, i.e., the amount of information about the aggregated neighbors of the target node considered. As shown in Figure 10, it can be seen that as the order of expansion increases, the performance of the framework

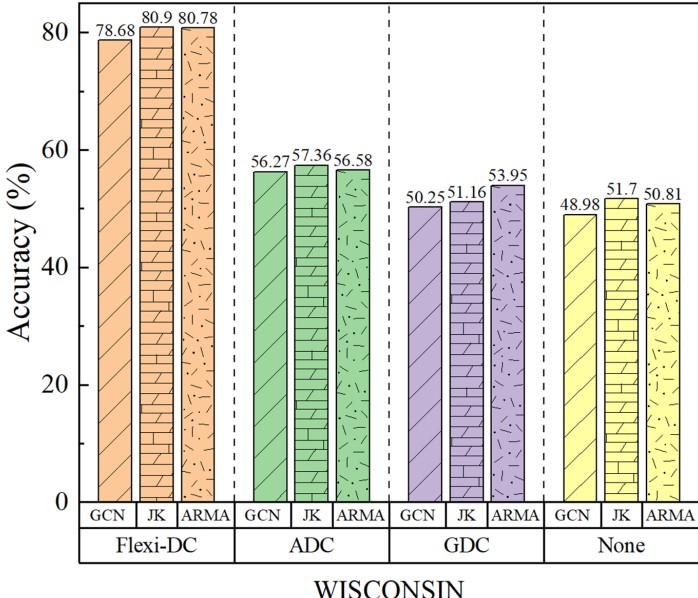

Figure 9: Algorithmic comparison of node classification accuracy on Cornell dataset, where JK refers to JKNet.

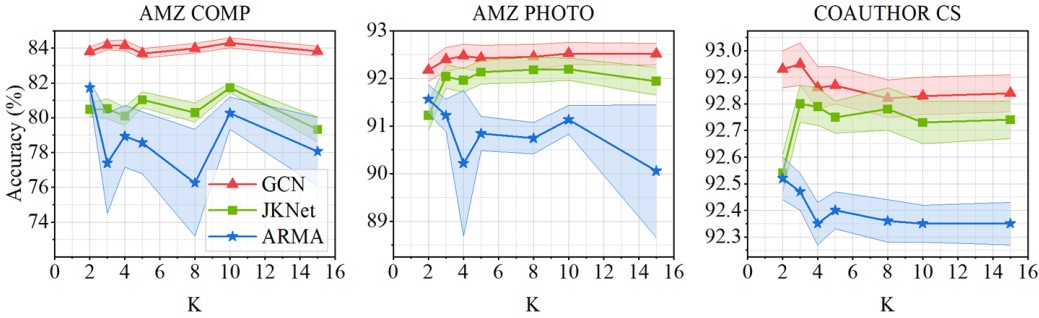

Figure 10: The influence of the Taylor expansion order (local neighborhood structure) on the performance of Flexi-DC. In most cases of homophilous graphs, increasing the expansion order helps to improve accuracy, while the opposite is true for heterophilous graphs.

initially improves but then basically remains unchanged and even begins to decrease. In addition, in Section 4.5, we also verified through ablation experiments that the diffusion module is less effective than the label smoothing module for large-scale datasets, which also proves that considering hidden information may lead to over-smoothing of the framework. This will be the main direction of our future research, exploring how to effectively combine the explicit and hidden information of nodes to improve their optimal vector representation on large-scale datasets while avoiding over-smoothing.

## C IMPLEMENT DETAILS

We further implemented flexible diffusion convolution and smooth labels on each backbone model. For flexible diffusion convolution, we set $k$ to be consistent with the number of Taylor expansion steps, enhancing the learning of hidden features of nodes within the neighborhood range. As for the smoothing strength, we set $\alpha$ to 0.8, which retains self-features while considering the features of neighboring nodes. We first aggregate information within the neighborhood range through node aggregation, followed by the diffusion and fully connected layer to obtain the predictive structure.

Table 4: The range of values for the hyperparameters.

| Symbols | Definitions |
|---|---|
| dropout | 0.5 |
| Hidden size | 64 |
| K_min | 1 |
| K_max | {2, 3, 4, 5, 8, 10, 15} |
| weight_decay | 0.01 |
| early_stopping | 100 |
| development | {1500, 5000, 120, 170} |
| optimizer | Adam |
| $\lambda$_init | 1 |
| $\sigma$ | Relu |

Finally, the smooth labels module corrects the predictions. Our architecture is designed with two layers. Similar to ADC and GDC, we only use their largest connected components. The data is split into a development and test set. Flexi-DC is a flexible component that can be directly integrated into existing GNN models, allowing them to flexibly learn hidden information of nodes within the neighborhood range and avoid over-confidence.

**Compute.** All the experiments are implemented with PyTorch, and tested on a machine with 32 Intel(R) Xeon(R) Platinum 8338C CPU @ 2.60GHz processors, one GPU of GeForce RTX 3090 with 24 GB memory size and fixed hyperparameter settings, which are described in the following.

**Hyperparameters.** For all experiments, the number of runs is set to 100, and the results of the runs are averaged with 95% confidence intervals via bootstrapping. The smoothing coefficient $\alpha$ is 1. The learning rate of $t$ is set to 0.01. The early stopping is set to patience of 100 epochs. Detailed hyperparameters can be found in Table 4

**Computational Complexity.** In Flexi-DC, the complexity of neighbor feature aggregation enhancement based on the degree of nodes is $\mathcal{O}(Nf\mathcal{N}_{max} + Nf\sum_{k=1}^{\mathcal{N}_i}\mathcal{N}_k)$, where $\mathcal{N}_{max}$ is the maximum neighbor structure hops, and $\mathcal{N}_i$ is the number of neighbor hops assigned by node $i$. After that, the model performs diffusion convolution, and the complexity is $\mathcal{O}(Nf^2c + NK_{max}c)$, where $K_{max}$ is here the diffusion step size, and $c$ is the channel. Finally, label smoothing is performed by degree-based neighbor assignment, and the complexity is $\mathcal{O}(N\mathcal{N}_{max} + N\sum_{k=1}^{\mathcal{N}_i}\mathcal{N}_k)$. Therefore, the computational complexity of Flexi-DC proposed in this paper is $\mathcal{O}(N((f+1)(\mathcal{N}_{max} + \sum_{k=1}^{\mathcal{N}_i}\mathcal{N}_k) + f^2c + K_{max}c))$.