# OpenReview forum: "Flexible Diffusion for Graph Neural Networks"
_ICLR.cc/2024/Conference — ICLR 2024 Conference Withdrawn Submission_

### Official Review · Reviewer_RJ7x · 2023-10-27

**Soundness:** 2 fair
**Presentation:** 1 poor
**Contribution:** 1 poor
**Rating:** 3
**Confidence:** 4

**Summary:**

The authors highlight GNN limitations: (1) fixed-range discrete message passing/aggregation and (2) insufficient adaptive aggregation and continuous diffusion, which affect performance. They then introduce Flexi-DC, which combines graph convolution with diffusion and label smoothing. Experiments shows that Flexi-DC elevates GNN performance across various datasets.

**Strengths:**

- The authors point out challenges in applying GNNs: (1) limited discrete message-passing range and (2) neglecting adaptive aggregation combined with continuous diffusion.
- The suggested Degree-based diffusion convolution offers a flexible and efficient continuous feature-passing approach.

**Weaknesses:**

- The presentation of this paper needs further justification.

    - Section 3 needs restructuring. The authors suggest integrating both spectral and spatial GNNs to enhance the model, but the selection and incorporation of these GNNs remain unclear. The Flexible Diffusion's handling of the input graph and its output embedding process lack clear elucidation. The supplementary code is also disorganized, making the whole model hard to interpret.

    - In section 3.1 there are multiple equations that relates to $\mathbf H^{(l+1)}_i$, which makes this section very confusing.

- The motivation appears insufficient. The authors' decision to integrate both spectral and spatial GNNs into the Flexible Diffusion for global information isn't well-justified. Additionally, the label smoothing module lacks clear rationale, and its relevance to the challenges highlighted by the authors is not evident.

- The innovation of Flexi-DC appears constrained. The Flexible Diffusion process, as a degree-related graph diffusion convolution, offers limited novelty. Furthermore, the label smoothing module is a familiar metric already employed by numerous other models.

- The experiments should be improved.

    - The experiment uses private random splits, even though the datasets selected by the authors have public splits available. I recommend the authors utilize the public splits to ensure a more accurate evaluation of their model in comparison to other baselines.
    - In the heterophily data mining domain, numerous baselines effectively address the issue. It's benifitial to include baselines like H2GCN [1], GCNII [2], CPGNN [3], and GPR-GNN [4] in the experiments. Comparing the proposed model to the state-of-the-art in this domain is essential for a thorough assessment.
    - Flexi-DC’s performance on several heterophily datasets (Cornell, Texas, Wisconsin) seems no better than simply applying an MLP to the feature matrices.

[1] Beyond Homophily in Graph Neural Networks: Current Limitations and Effective Designs

[2] Simple and Deep Graph Convolutional Networks

[3] Graph Neural Networks with Heterophily

[4] Adaptive Universal Generalized PageRank Graph Neural Network

**Questions:**

While the authors tout Flexi-DC's efficiency, there's no clear comparison of its running time against GCN and other baselines highlighted in the Weakness section. How does it fare in terms of computational speed relative to these models?

---

### Official Review · Reviewer_co1S · 2023-10-29

**Soundness:** 1 poor
**Presentation:** 2 fair
**Contribution:** 2 fair
**Rating:** 3
**Confidence:** 3

**Summary:**

This paper proposes a flexible aggregation mechanism that adapts to different node degrees and local structures of the nodes. The proposed method achieves superior performances on many graph datasets.

**Strengths:**

The authors provide a very detailed experiment design, which support the performance improvement of the proposed method.

**Weaknesses:**

In my opinion, the motivation of this paper, that discrete aggregation mechanism is a problem, needs to be further explained. This hinders me from acknowledging the necessity of the continuous propagation. To be detail, the following expressions are unclear and confusing.

1. For example, only considering the aggregation of direct neighbors is inconsistent with the structure of graph data, and the discrete aggregation operation limits the information passing between nodes … and there is strong evidence that these two limitations can restrict the expressive power and applicability of GNNs (the first paragraph of the Introduction).
Q: What does “…inconsistent with the structure of graph data …” mean? Why does the discrete aggregation operation limits the information passing? And where is the evidence? These need to be further explained.

2.  I cannot understand why “The former (degree-based diffusion convolution) leads to more flexible and smooth feature passing, thus improving the expressiveness and applicability of the model” (The 2nd point of Summary of Contributions on P2). Also, “while the latter allows the framework to avoid over-confidence”. The definition of “over-confidence” needs to be clarified.

3. “GNNs typically rely on discrete feature aggregation and propagation operations to update node information, which can lead to loss of embedding information and accuracy” (the last paragraph on P2) ——The authors still do not explain why “discrete” aggregation lead to the results. After all, the solution Eq. (2) is also in a discrete form. It would be more persuasive if the authors argue that the problem is the finite receptive field.

4. The full Intuition part on P3-4 is not convincing. First, what does “size of local structure” mean? The hop or the number of nodes? The expression “larger local structures may lead to over-smoothing, while smaller ones may result in instability and insufficient information aggregation” needs to be clarified, as well as “the local structure is also discrete, making it impossible to describe the continuous relationship between nodes. This limits the information propagation between nodes and may result in information loss or inaccuracies”. What is the “continuous relationship between nodes”?

5. “If the traditional GNNs aggregate only the immediate neighbors (1-hop) for message passing, the nodes with higher degrees receive too much information, causing over-smoothing, whereas the nodes with lower degrees may not have enough information for self-enhancement.” (The first paragraph of Sec 3.1 on P4). It is not strict to say too much information causes over-smoothing without evidence.

Finally, the authors argue that “This approach mitigates the limitation of considering only immediate neighbors in previous methods and effectively avoids issues such as $\textbf{over-smoothing}$ and insufficient information aggregation” (In Conclusion). However, I cannot find any experiments supporting this.

**Questions:**

Please refer to the Weakness part. I'd be happy to change my opinion if given more clarifications.

---

### Official Review · Reviewer_JgmZ · 2023-10-30

**Soundness:** 2 fair
**Presentation:** 2 fair
**Contribution:** 2 fair
**Rating:** 3
**Confidence:** 4

**Summary:**

The paper introduces "Flexible Diffusion for Graph Neural Networks," which addresses the limitations of traditional GNNs in modeling graph-structured data. The authors propose a novel framework called Flexi-DC, which combines degree-based diffusion convolution and label smoothing to overcome limitations related to node degree and local structure. Flexi-DC outperforms vanilla GNN implementations and existing state-of-the-art methods across various real-world datasets, particularly demonstrating its efficiency on large-scale data. This approach has the potential to significantly enhance the applicability of GNNs in diverse applications involving graph-structured data.

**Strengths:**

1.	Novel Contribution: The paper introduces a novel framework, Flexible Diffusion Convolution (Flexi-DC), that addresses the limitations of existing GNNs by incorporating degree-based diffusion convolution and label smoothing.
2.	Application in Various GNN Models: The paper highlights the versatility of Flexi-DC by demonstrating its applicability to a variety of GNN models, indicating its potential to enhance the performance of different GNN architectures.

**Weaknesses:**

1. Lack of Experiments: Although the authors emphasize that Flexi-DC works on both homophily and heterophily datasets, the experiments are only conducted on the Cornell, Wisconsin, and Texas datasets, all of which are relatively small, with fewer than 200 nodes. This raises concerns about its performance on larger-scale heterophily datasets [1][2].

2. Lack of Theoretical Proof: The paper states that "nodes with higher degrees receive too much information, causing over-smoothing, whereas nodes with lower degrees may not have enough information for self-enhancement." However, this statement is based solely on intuition. The inclusion of theoretical proof would better justify this claim and the overall motivation of the paper.

3. Equation: $S^{\prime}=\sum_{k=0}^K \theta_k \mathbf{U} T^k(\tilde{\mathbf{\Lambda}}) \mathbf{U}^T S=\sum_{k=0}^K \theta_k T^k(\tilde{\mathbf{L}}) S$ represents the Chebyshev K-order truncation used to approximate the convolution kernel. It shares a similar form with other spectral methods, such as ChebNet, BernNet, ChebNetII [5], GPRGNN [3], and JacobiConv [6]. I am curious about the distinctions between Flexi-DC and other spectral methods. It appears that, despite the authors claiming novelty, Flexi-DC converges towards conventional spectral methods. Furthermore, label smoothing is not a novel concept, as it has been proposed and utilized in methods like APPNP[4] and GPRGNN [3].

4. Lack of Baselines: The paper presents the performance of Flexi-DC when applied to GCN, JKNet, and ARMA. However, its compatibility with state-of-the-art (SOTA) baselines is not explored. For example, it would be beneficial to compare Flexi-DC+(GCN/ARMA/JKNet) with other SOTA methods mentioned earlier. If it outperforms these methods, it would enhance its practicality significantly.




Reference:
1.	Graph Neural Networks for Graphs with Heterophily: A Survey.
2.	Large Scale Learning on Non-Homophilous Graphs: New Benchmarks and Strong Simple Methods.
3.	ADAPTIVE UNIVERSAL GENERALIZED PAGERANK GRAPH NEURAL NETWORK.
4.	Predict then Propagate: Graph Neural Networks meet Personalized PageRank.
5.	Convolutional Neural Networks on Graphs with Chebyshev Approximation, Revisited.
6.	How Powerful are Spectral Graph Neural Networks?

**Questions:**

See Weaknesses.

---

### Official Review · Reviewer_f45a · 2023-10-30

**Soundness:** 2 fair
**Presentation:** 1 poor
**Contribution:** 2 fair
**Rating:** 3
**Confidence:** 4

**Summary:**

This work aims to address the limitation of existing graph neural networks (GNNs), that only fixed-range discrete message-passing operations can be performed.
To tackle it, the authors utilized diffusion-based GNNs.
However, diffusion-based GNNs usually perform fixed $K$-hop aggregation for all nodes, leading to large differences in the size of receptive fields for different nodes.
The authors claim that aggregation with a large receptive field might lead to oversmooshing while with a small receptive field might lead to instability. Thus, they propose to use the node-degree to control the effect receptive field adaptively for each node, i.e., different nodes have different $K$-hop aggregation based on node-degree.
To reach a better performance, the proposed method also equips with bi-level optimization and label smoothing techniques during training.
Evaluated on several node-level benchmarks, the proposed method Flexi-DC outperforms previous diffusion-based MPNNs on smaller datasets and reaches comparable performance on larger datasets with fewer computations.

**Strengths:**

1. This work proposed a novel hypothesis that one should balance the sizes of receptive fields for different nodes, in order to reach a global balance between over-smoothing and instability, potentially led by too large/small receptive fields.
The experiment can support it empirically in general.
2. The figures comparing different techniques on different datasets are really good: clear and straightforward.

**Weaknesses:**

1. Even though I can generally understand the proposed method, the presentation is not clear and even confusing, especially on the math formulas;
    The notation is really confusing:
    - In eq.1 $h_i^{(l)}$ is defined as the hidden representation of node $v_i$ at layer $l$, which is supposed to be a vector. But in section 3.1, $\mathbf{H}\_i^{(l)}$ seems to be the same as $h\_i^{(l)}$ (mentioned  in the context as *the features of the target node*). If so, the equation in the first paragraph of sec.3.1 seems problematic (only node $i$ is involved).
     -  $\mathbf{Kel}\_\beta$ is not defined before eq. 3 and eq. 4; It's hard to tell whether it is a scalar, vector, matrix, or a function.
    From eq. 4,  ${\mathbf{Kel}\_{\beta\_i}^k}  := \frac{\lambda^k e^{-\lambda}}{k!}$, which is a scalar independent on $i$.
    And eq. 4 just performs a summation of  the same $\mathbf{H}^{(l)}\_{i}$ with different  $k$  to get  $\mathbf{H}\_{i}^{(l+1)}$.
    - In the formula for *well-known continuous Laplacian*, the $x$ is not defined or mentioned before.
    - If I understand correctly, $K$ shall be a function of node degree (as given by the first inline equation in sec. 3.1). But it is not shown in eq. 4.

2. The authors claim that graph diffusion extends the "discrete" message-passing to a "continuous" form. Personally, I doubt whether this claim is still valid for a small value of $K$ (i.e., bounded by node degrees in this work), since $k$ needs to be an integer in eq. 2 and the approximation $\sum_{k=1}^\infty \mathbf{T}^k \approx (\mathbf{I} - \mathbf{T})^{-1}$ is only valid when $k \to \infty$ (Xhonneux et al., 2020).

3. The bi-level optimization strategy and label smoothing techniques can improve the performance of the proposed Flexi-DC. However, these two techniques can be directly introduced to previous techniques (such as GDC).
In the current experimental setting, it is hard to tell whether the outperformance is contributed by the proposed Flexi-DC components or the bi-level optimization strategy, given the fact that the gap is more remarkable on smaller datasets than on some larger datasets.
An additional comparison such as *Flexi-DC w/o bi-level* or *GDC/ADC w/ bi-level* will be very helpful to better support the technique.

**Questions:**

1. As listed in the weaknesses

2. What is the definition of *Flexible-DC w/o Diffusion* in the ablation study? The *Flexible* is defined together with diffusion in methodology part. It is unclear how to be *Flexible* without *Diffusion*.

3. *Flexi-DC* is only evaluated on node-level tasks. Is the hypothesis (about the size of receptive fields) also valid for graph-level tasks? The scope of the proposed techniques might need to be clarified.